# Classification of Vascular Plants in Vietnam According to Modern Classification Systems

**DOI:** 10.3390/plants12040967

**Published:** 2023-02-20

**Authors:** Ngoc A. Luu-dam, Ngan T. Lu, Thai H. Pham, Truong V. Do

**Affiliations:** 1Vietnam National Museum of Nature, Vietnam Academy of Science and Technology, 18th Hoang Quoc Viet Road, Cau Giay, Hanoi 100000, Vietnam; 2Graduate University of Science and Technology, Vietnam Academy of Science and Technology, 18th Hoang Quoc Viet Road, Cau Giay, Hanoi 100000, Vietnam

**Keywords:** angiosperms, APG IV, classification, gymnosperms, lycophytes and ferns, PPG1, vascular plants, Vietnam

## Abstract

Vietnam is extremely rich in biodiversity, with a remarkable range of habitats and more than 13,500 species of vascular plants recorded for the flora of Vietnam. This number represents about 3 to 5% of the world’s diversity of vascular plants. Over the past 30 years, there were two important documents on the vascular plants of Vietnam published, *An Illustrated Flora of Vietnam* (IFV) and *Checklist of Plant Species of Vietnam* (CPSV). During the past half century, the advent of molecular phylogenetics has witnessed dramatic changes in the classifications of vascular plants, and some modern classification systems of vascular plants have been established, e.g., PPG I, GPG, and APG. However, the vascular plants of Vietnam have not yet been classified according to these modern classification systems. In this paper, we present the history of the classification of vascular plants in Vietnam, compare the circumscription of all families of vascular plants occurring within Vietnam in IFV, CPSV, and the modern classification systems when applicable, and summarize familial assignments of all controversial genera in the different classifications. Furthermore, we also arrange the 37 families of lycophytes and ferns occurring within Vietnam according to the latest classification system (PPG I) and the 8 families of gymnosperms according to the latest Christenhusz’s system (GPG). The 246 families of angiosperms are arranged according to the fourth edition of the latest Angiosperm Phylogeny Group (APG IV). These results are the foundation stones and would be helpful for future research on the flora of Vietnam and the arrangement of plant collections in Vietnamese herbaria based on the updated classifications.

## 1. Introduction

Vietnam is one of the most biodiverse countries, with a huge variety of distinctive and fascinating wildlife. It is adjacent with the natural geography of southern China, where the diversity of extant vascular plants is extraordinary [1]. The flora of Vietnam is also rich in species composition, with more than 20,000 plant species recorded [2], containing many endemic taxa and key and primitive taxa of angiosperm phylogeny, possibly the origin of East Asian flora, and even angiosperms [3]. Therefore, Vietnam is a significant bioregional area in terms of physical geography and biogeography, necessitating biodiversity conservation.

Biological classifications are fundamental tools for communication about biodiversity [4]. The stability of names is thus of great importance, and it is critical to consider existing classifications when determining taxa worthy of recognition and the ranks at which to treat them. A focus on natural groups is similarly essential, as it results in classifications that reflect the evolutionary history and ultimately lead to greater stability. Nowadays, the surge in the application of molecular biology techniques has resulted in significant contributions to plant systematics, evolutionary questions, phylogeny, and the studies of plant diversity. Primarily, the molecular biology techniques have obtained stimulating achievements in plant systematics. As a result, various DNA-based modern classification systems were established, e.g., APG (the Angiosperm Phylogeny Group classification) [5], APG II [6], APG III [7], APG IV [8] for flowering plants, PPG I (the Pteridophyte Phylogeny Group classification) [9] for lycophytes and ferns, and GPG (the Gymnosperm Phylogeny Group classification) [10] for gymnosperms. Together, these are the modern, comprehensive classifications for vascular plants (lycophytes and ferns, gymnosperms, and angiosperms). These classifications present the linear sequence and delimitation of genera and families of vascular plants, recognizing their natural groups based on both morphological and molecular data analyses, utilizing a democratic community-based approach, which is currently widely accepted by scientists and herbaria around the world. These classifications are valued for their predictive value. Modern molecular methods are the latest tools in our successive approximation to recognize past superficial similarities and convergent evolution and try to work out the evolutionary history of plants [1].

Vascular plants of Vietnam have been documented since the beginning of the 18th century by French botanists [11,12,13] and domestic botanists [2,14,15,16,17,18]. The resultant texts *An Illustrated Flora of Vietnam* (IFV) [17] and *Checklist of Plant Species of Vietnam* (CPSV) [2] recognized, classified, and enumerated most species, genera, and families of vascular plants in Vietnam. 

These two documents are very useful and commonly used in Vietnam. However, the linear sequence and delimitation of recognized genera and families of vascular plants in Vietnam in these previous treatments were according to classical classification systems, which were established mainly based on morphological data that did not reflect evolutionary history and, ultimately, led to the instability of natural groups (genera and families).

Furthermore, there was no update of the modern classifications for vascular plants (lycophytes and ferns, gymnosperms, and angiosperms) in Vietnam, so the genera and families of vascular plants of Vietnam have not yet been classified and arranged according to modern classification systems, even though the volumes on angiosperm families in the *Flora of Vietnam* were recently published after the publication of APG IV [8].

In this paper, we present the history of the classification of vascular plants in Vietnam; compare the circumscription of all families of vascular plants occurring within Vietnam in IFV, CPSV, and the modern classification systems (PPG I, GPG, and APG IV) [8,9,10], when applicable in the form of tables; and summarize the familial assignments of all controversial genera in the different classifications. Furthermore, the results not only present the history of the studies of vascular plants in Vietnam, but also reflect the transition from early morphology-based classifications to modern DNA sequence data-based classifications. These results are the foundation stones and would be helpful for future research on the flora of Vietnam and the arrangement of plant collections in Vietnamese herbaria based on the updated classifications.

## 2. Results and Discussion

### 2.1. History of Classification of Vascular Plants in Vietnam

Vascular plant species of Vietnam have been initially documented since the beginning of the 18th century by French botanists from the Museum National d’Histoire Naturelle, Paris. They began with the flora of the southern parts of the country. João de Loureiro (1710–1791) initially studied and wrote the first one, *Flora Cochinchinensis* (in Latin), including Vols. 1–2 [11]. Loureiro’s work was an extension of his interest in local medicinal plants, in the service of the King of Cochinchina, a region in the southern third of modern Vietnam that included Ho Chi Minh city and Thua Thien Hue province, and at some later point, a French colony. Loureiro’s specimen collections, which would have been important reference material, and his plant descriptions were very brief. His work was followed by Pierre Louis (1833–1905), who published 26 parts bound in 4 volumes of *Flore forestière de la Cochinchine* (in French) [19] Later, Francois Gagnepain (1866–1952), Henri Lecomte (1856–1934), and Henri Humbert (1887–1967) published *Flore générale de l’Indo-Chine* (FGIC), comprising Vols. 1–7, for the three countries of the Indochinese region: Vietnam, Laos, and Cambodia [12]. The families in *Flore générale de l’Indo-Chine* were classified according to “*Le Genera Plantarum*” [20]. 

While revising the previous treatments for the floras of Vietnam, Laos, and Cambodia, plant taxonomists from the Museum National d’Histoire Naturelle, Paris, and domestic taxonomists [11,12,19] compiled *Flore du Cambodge, du Laos et du Viêtnam* in French (FCLV), a project begun by André Aubréville in 1960 to succeed *Flore générale de l’Indochine*, coordinated by H. Lecomte and H. Humbert during 1907–1934, and H. Humbert until 1951 [12]. That last issue was Vol. 32 in 2004 for Myrsinaceae. The whole project consists of 32 volumes, dealing with 1552 species in 77 families of vascular plants [13], of which 69 families in 31 volumes are angiosperms; only 1 volume (Vol. 28) is for 8 families of the gymnosperms. None of the fern and lycophyte families were treated. 

Since 2013, the series has been renewed in the English-language version as *The Flora of Cambodia, Laos, and Vietnam* and jointly co-published by the Museum National d’Histoire Naturelle, Paris, and the Royal Botanic Garden of Edinburgh. This book series is considered to be an ongoing project. The most recently published volumes were Vol. 33 for Apocynaceae [21], Vol. 34 for Polygonaceae [22], Vol. 35 for Solanaceae [23], and Vol. 36 for Convolvulaceae [24]. Several volumes are still in revision, and more volumes will come out in the future. So far, not many species and families are recorded in each volume, far from the total number of vascular plants in these three countries in general and Vietnam in particular.

The Vietnamese-language *Flora of Vietnam* project (FOV) was initiated in 1996 by domestic taxonomists from the Institute of Ecology and Biological Resources, Vietnam Academy of Science and Technology, and Hanoi National University, Vietnam, which is a scientific effort to publish the account of the species of vascular plants of Vietnam, and the first volume (Vol. 1) was published in 2000 for Annonaceae. The taxonomic concepts and delimitation of the families in FOV mainly followed the classical classification systems, such as Bentham et Hooker [20], Crosquist [25], Melchior [26], and Takhtajan [27,28,29]. The FOV project took nearly 20 years of extraordinary effort by 2 generations of Vietnamese botanists, with a total of 21 individuals participating. The FOV project published 21 volumes, which are subdivided into 2 periods. The initial period, which lasted from 1996 to 2007, with 11 volumes of 30 angiosperm families and 1 phylum and 1 order of green algae published, i.e., Vol. 1—Annonacae [30], Vol. 2—Lamiaceae [31], Vol. 3—Cyperaceae [32], Vol. 4—Myrsinaceae [33], Vol. 5—Apocynaceae [34], Vol. 6—Verbenaceae [35], Vol. 7—Asteraceae [36], Vol. 8—Liliales [37], Vol. 9—Orchidaceae-*Dendrobium* [38], Vol. 10—Chlorophyta [39], and Vol. 11—Fucales and Polygonaceae [40].

The later period was from 2008 to 2017, with 10 volumes of 11 angiosperm families published, i.e., Vol. 12—Spindaceae [41], Vol. 13—Arecaceae [42], Vol. 14—Malvaceae [43], Vol. 15—Asclepidaceae [44], Vol. 16—Araceae [45], Vol. 17—Solanaceae & Loganiaceae [46], Vol. 18—Gesneriaceae [47], Vol. 19—Theaceae [48], Vol. 20—Lauraceae [49], and Vol. 21—Zingiberaceae [50].

None of the families of the ferns, lycophytes, and gymnosperms have been published in the FOV. Therefore, representation of the entirety of flora is still lacking in coverage of the total number of vascular plant families in the country. 

Pham Hoang Ho (1929–2017) originally published *Illustrated Flora of Southern Vietnam* with the support of the South Vietnamese Ministry of Education in 1969 [14]. Its second edition was published in two volumes in 1970 and 1972, respectively. Later, more plant species from Northern Vietnam were added to the last edition [15]. Therefore, it was revised in a new version, *An Illustrated Flora of Vietnam* (Cây cỏ Việt Nam) (IFV). The first edition included three volumes in Vietnamese [16]. The last edition also published 3 volumes in Vietnamese, which documented 11,611 species, 2249 genera, and 284 families of vascular plants, of which 139 genera and 33 families are ferns and lycophytes, 25 genera and 10 families are gymnosperms, and the rest are genera and families of angiosperms [17]. Each species includes nomenclature, a brief description, and a simple line drawing. The book is currently relatively complete, so the usage rate is relatively high by both foreign and Vietnamese botanists to identify plants. Although the classification system and delimitation of families and genera of vascular plants in the book were not mentioned, they were probably adopted from the previous treatments by French botanists [11,12,13].

Based on various published documents in Vietnam, as well as neighboring countries, such as China, Thailand, Laos, and Cambodia, and non-published materials by Vietnamese, botanists completed a *Checklist of Plant Species of Vietnam* (CPSV). The working group was comprised of more than 40 botanists from the Institute of Ecology and Biological Resource (IEBR), Vietnam Academy of Science and Technology, Center for Natural Resources and Environmental Studies (CRES), and Vietnam National University in Hanoi, in collaboration with some Russian botanists. The completed *Checklist of Plant Species of Vietnam* (CPSV) from Procaryophyta, Fungi, and Algae to Angiospermae, was divided into three volumes. The first volume (Vol. 1) is composed of phylum from Procaryota to Gymnospermae and part of angiosperms, while the two other volumes (Vols. 2 & 3) are exclusively for the rest of the Angiospermae in Vietnam [2]. For the angiosperms, the delimitation of the families was mainly adopted from previous work [18], which classified the angiosperms in Vietnam following Takhtajan’s classification systems [28,29]. Species are assigned to genera and families according to the Kew Herbarium Concept [51]. By family, each species includes full nomenclature and a brief description of morphology, distribution, some ecological features, and their use or harm. These three volumes documented 11,550 species and subspecies of 2374 genera, and 308 families of vascular plants in Vietnam. Of this total, 700 species and subspecies of 143 genera and 34 families are ferns and lycophytes, 70 species, and subspecies of 21 genera and 9 families are gymnosperms, and 10,780 species and subspecies of 2210 genera and 265 families are Angiospermae.

### 2.2. Classification of Vascular Plants of Vietnam according to Modern Classification Systems

The comparison of circumscription and classification of vascular plants of Vietnam in IFV, CPSV, and PPG 1 (for lycophytes and ferns), GPG (for gymnosperms), and APG IV (for angiosperms), is presented in the seven following tables (please see the Appendix A).

#### 2.2.1. Lycophytes and Ferns

For the lycophytes and ferns of Vietnam (Appendix A), IFV (Vol. 1) recognized 139 genera and 33 families, while CPSV (Vol. 1) recognized 143 genera and 34 families. The delimitation of genera and families in both IFV and CPSV is the same, with 28 out of the 33 families of lycophytes and ferns in IFV having the exact delimitation as CPSV (Appendix A). These 28 families are morphologically homogenous, and the circumscription of these families has rarely been controversial [9,52].

However, the delimitation of some genera and families is different between IFV and CPSV. The family Oleandraceae (incl. *Arthropteris, Nephrolepis,* and *Oleandra*) sensu CPSV was not recognized in IFV. The only member of Thyrsopteridaceae (*Cibotium*) sensu IFV is accommodated in Dicksoniaceae in CPSV; *Angiopteris* and *Archangiopteris* (Angiopteridaceae) sensu IFV were transferred to Marattiaceae in CPSV; and some genera (*Athyrium, Cystopteris, Diplazium*) of Aspleniaceae sensu IFV were transferred to Woodsiaceae in CPSV, which was not recognized in IFV.

Based on the previous study [52], Phan [53] classified all species of ferns from Vietnam, but not including lycophytes, into 135 genera, 28 families, 11 orders, and 4 classes, of which 3 families (Lindsaeaceae, Cibotaceae, and Lygodiaceae) are newly circumscribed for Vietnam (Appendix A). The results showed that Ophioglossaceae belonging to Polypodiophyta was transferred to another class, Psilotopsida; especially Marattiaceae belonging to Polypodiophyta was split into its own class, namely, Marattiopsida. Three families sensu IFV and CPSV were merged with other families in Phan, suggesting that the circumscription of these three families is broader than the corresponding families recognized in IFV and CPSV. Indeed, based on chloroplast sequence data, some families, sensu IFV and CPSV, were merged in Phan, and are apparently monophyletic, but nested within other families, e.g., Azollaceae (in Salviniaceae [54]) and Cheiropleuriaceae (in Dipteridaceae [55]).

According to PPG I [9], the lycophytes and ferns of Vietnam were classified into 2 classes, 3 sub-classes, 14 orders, 37 families, and 134 genera. By following this modern classification, the families of lycophytes and ferns in Vietnam are more than the previous treatments [2,17,53], of which six families (Cystopteridaceae, Rhachidosaraceae, Diplaziopsidaceae, Didymochlaenaceae, Hypodematiaceae, and Nephrolepidaceae) were newly circumscribed for *Flora of Vietnam*, but its number of genera is less than those of IFV, CPSV, and Phan (Appendix A; Figure 1).

The delimitation of families and genera of lycophytes and ferns in Vietnam according to IFV and CPSV is significantly different from those of PPG I, while Phan’s treatment is mostly the same as PPG I. Some minor changes come from *Acystopteris*, *Rhachidosorus*, and *Diplaziopsis* (Woodsiaceae in Phan), which were transferred to Cystopteridaceae, Rhachidosoraceae, and Diplaziopsidaceae, respectively; *Athyrium*, *Deparia*, and *Diplazium* (Woodsiaceae in CPSV and Phan) were transferred to Athyriaceae; *Didymochlaena* (Dryopteridaceae in IFV, CPSV, and Phan) was transferred to Didymochlaenaceae; *Hypodematium* (Woodsiaceae in CPSV or Dryopteridaceae in Phan) and *Leucostegia* (Davalliaceae in IFV, CPSV, or Dryopteridaceae in Phan) were transferred to Hypodematiaceae; *Nephrolepis* (Davalliaceae in IFV or Oleandraceae in CPSV or Lomariopsidaceae in Phan) was transferred to Nephrolepidaceae (Appendix A).

Furthermore, the delimitation of some genera changed. Phan [53] recognized two genera, *Caobangia* and *Kontumia* (Polypodiaceae), endemic to Vietnam, but APG 1 merged them into *Lemmaphyllum* and *Leptochilus,* respectively, which were in line with the molecular-based phylogenetic studies [56,57].

#### 2.2.2. Gymnosperms

For the gymnosperms of Vietnam (Appendix A), IFV (Vol. 1) recognized 25 genera and 10 families, while CPSV (Vol. 1) identified 22 genera and 9 families, of which Araucariaceae was not recognized in CPSV. The delimitation of genera and families of gymnosperms in Vietnam in both IFV and CPSV is the same, with eight out of the nine families of gymnosperms in IFV having the exact same delimitation as CPSV (Appendix A). The significant difference between IFV and CPSV comes from the delimitation of *Amentotaxus*, which was placed in Amentotaxaceae in IFV, while CPSV transferred it to Taxaceae (Appendix A). The recognition of *Amentotaxus* in Taxaceae is in line with molecular study [58].

In the book *Conifers of Vietnam*, Nguyen & Thomas [59] presented 33 native conifer species of Vietnam, which were classified into 19 genera and 5 families, of which 3 previously recognized genera, i.e., *Pseudotsuga* (Pinaceae), *Taiwania* and *Xanthocyparis* (Cupressaceae), were added to the gymnosperms in Vietnam, but transferred *Cunninghamia* and *Glyptostrobus* (in Taxodiaceae sensu IFV and CPSV) to Cupressaceae based on the previous study [60], which merged Taxodiaceae into Cupressaceae.

According to GPG [10], the gymnosperms of Vietnam were classified into 4 sub-classes, 6 orders, 8 families, and 23 genera (Appendix A). By following this modern classification, the families of gymnosperms in Vietnam are less than the previous treatments [2,17] (Appendix A, Figure 2), of which nine genera in IFV, CPSV, and Nguyen & Thomas [2,17,59] have the exact delimitation with those in GPG (Appendix A). The delimitation of Cupressaceae and Taxaceae sensu IFV, CPSV, and Nguyen & Thomas has remarkable changes in comparison with those in GPG, of which *Glyptostrobus* and *Cunninghamia* (Taxodiaceae) were transferred to Cupressaceae; *Xanthocyparis* (Cupressaceae)*,* a genus endemic to Vietnam [60], was assigned to be a synonym for *Cunninghamia* (Cupressaceae)*;* while *Cephalotaxus* (Cephalotaxaceae) and *Amentotaxus* (Amentotaxaceae) were transferred to Taxaceae.

#### 2.2.3. Angiosperms

For the angiosperms of Vietnam (Appendix A), IFV (Vols. 1 & 3) recognized 2110 genera and 241 families. In contrast, CPSV recognized 2210 genera, 265 families (not including 14 families belonging to Liliaceae s.l.), and 2 classes (Magnoliopsida and Liliopsida), of which ca. 1645 genera and 219 families are Magnoliopsida (eudicots), and ca. 565 genera and 46 families are Liliopsida (monocots) (Appendix A, Figure 3). However, both IFV and CPSV did not classify these families of angiosperms from Vietnam into orders. Until now, CPSV has been considered the complete enumeration of the vascular plants in Vietnam. Of the 265 families of angiosperms in Vietnam, there are 234 families recognized in both IFV and CPSV. Furthermore, two other families (Zannichelliaceae and Butomaceae) in IFV were not identified in CPSV. Still, the delimitation of these two families is the same as those of Cymodoceae and Limnocharitaceae in CPSV.

According to CPSV, the number of families of angiosperms in Vietnam was significantly raised, with 31 families newly circumscribed for Flora of Vietnam, i.e., Calycanthaceae, Altingiaceae, Molluginaceae, Tetragoniaceae, Bonnetiaceae, Hypericaceae, Pyrolaceae, Diapensiaceae, Escalloniaceae, Hydrangeaceae, Penthoraceae, Chrysobalanaceae, Hugoniaceae, Davidiaceae, Aucubaceae, Mastixiaceae, Toricelliaceae, Helwingiaceae, Sịphonodontataceae, Schoepfiaceae, Erythropalaceae, Dipentodontaceae, Viscaceae, Potaliaceae, Cuscutaceae, Martyniaceae, Lobeliaceae, Zosteraceae, Hypoxidaceae, Asparagaceae, and Costaceae (Appendix A). Furthermore, Liliaceae s.l. sensu CPSV was separated into 14 families, i.e., Agavaceae, Alliaceae, Amaryllidaceae, Asphodelaceae, Asteliaceae, Convallariaceae, Dracaenaceae, Hemerocallidaceae, Hyacinthaceae, Liliaceae, Melanthiaceae, Nolinaceae, Phormiaceae, and Trilliaceae, which are included within closely related families, of which three families (Agavaceae, Amaryllidaceae, and Liliaceae s.s.) are also recognized in IFV (Appendix A). 

According to APG IV [8], the angiosperms of Vietnam were classified into 246 families and 54 orders of 6 groups (basal angiosperms, magnoliids, Chlorantheles, monocots, Ceratophyllales, and eudicots). The number of families is more than those of IFV (238 families), but the number of families is less than those of CPSV (265 families). Of these, 203 and 225 families were also recognized in IFV and CPSV, respectively. Of the 199 families identified in IFV, CPSV, and APG IV, 133 families in IFV and CPSV have the exact delimitation as in the APG IV classification (Appendix A).

Linear classification of angiosperms in Vietnam following APG IV is presented in Appendix A. Below are summaries of classifications of basal angiosperms, magnoliids, Chlorantheles, monocots, Ceratophyllales, and eudicots in IFV, CPSV, and APG IV. 

*Basal angiosperms*: IFV and CPSV recognized five families (Cabombaceae, Nymphaeaceae, Barclayaceae, Ceratophyllaceae, and Nulumbonaceae) in the basal angiosperms, which include Amborellales, Nymphaeales, and Austrobaileyales (called as the ANA group), of which only Cabombaceae sensu IFV and CPSV has the exact delimitation as APG IV, while Barclayaceae was placed in Nymphaeaceae and Ceratophyllaceae (Ceratophyllales) was placed in the independent order in APG IV, which is the probable sister of eudicots. Nulumbonaceae was placed in the eudicot order Proteales. Furthermore, Illiciaceae and Schisandraceae were placed in the eudicot (Magnoliopsida) sensu IFV and CPSV and transferred to the order of Austrobaileyales in APG IV. Therefore, the basal angiosperms in Vietnam include two orders and three families, according to APG IV. The order Austrobaileyales has not yet been circumscribed for Flora of Vietnam from the previous treatments (Appendix A). 

*Magnoliids*: The classification of magnoliids in Vietnam has been less controversial. The same nine families of magnoliids of Vietnam sensu IFV were also recognized in CPSV and APG IV, i.e., Saururaceae, Piperaceae, Aristolochiaceae, Myristicaceae, Magnoliaceae, Annonaceae, Hernandiaceae, Monimiaceae, and Lauraceae. However, Calycanthaceae was not recognized in IFV, but identified in both CPSV and APG IV. The delimitation of Magnoliales sensu CPSV and APG IV is the same, which includes Myristicaceae, Magnoliaceae, and Annonaceae. In contrast, the delimitation of some other families sensu CPSV is different from those of APG IV. Aristolochiaceae and Rafflesiaceae sensu CPSV were placed in Aristolochiales. However, APG IV placed Aristolochiaceae, together with Saururaceae and Piperaceae, in Piperales, and Rafflesiaceae was placed in Malpighiales (Eudicots). Laurales sensu CPSV includes five families, i.e., Calycantaceae, Hernandiaceae, Monimiaceae, Lauraceae, and Chloranthaceae, which are the same as APG IV, except for Chloranthaceae, which is an independent lineage and placed in the own order Chloranthales. Therefore, the magnoliids in Vietnam include ten families and three orders, according to APG IV (Appendix A).

*Chloranthales*: Only one family, Chloranthaceae, was recognized. Wickett et al. [61] did not support the sister group with Magnoliids. Therefore, APG IV placed the order as an independent lineage with the remaining lineages (Magnoliids and Eudicots/Monocots/Ceratophyllaceae). However, Chloranthaceae sensu IFV and CPSV were placed in the order Laurales.

*Monocots*: The delimitation of the families and orders in the three classifications (IFV, CPSV, and APG IV) is significantly different. IFV recognized 45 families in Monocots, while CPSV recognized 58 families, not including 14 separated from Liliaceae sensu IFV (Appendix A). The families Nolinaceae, Hyacinthaceae, Dracaenaceae, Convallariaceae, and Agavaceae were merged into Asparagaceae, while Hemerocallidaceae and Phormiaceae were merged into Asphodelaceae. Furthermore, Melanthiaceae sensu CPSV was divided into Colchicaceae (incl. *Gloriosa* and *Iphigenia*) and Petrosaviaceae (*Petrosavia*) in APG IV, while Melanthiaceae sensu APG IV includes Trilliaceae and Liliaceae (*Paris*). Furthermore, 27 of 57 families sensu IFV and CPSV have the same delimitation as APG IV. Some families sensu IFV and CPSV were merged into the other families in APG IV, such as Najadaceae was merged into Hydrocharitaceae; Zannichelliaceae in IFV => Cymodoceaceae; Trilliaceae in CPSV => Melanthiaceae; Phormiaceae and Hemerocallidaceae in CPSV => Asphodelaceae; Alliaceae in CPSV => Amaryllidaceae; Dracaenaceae and Hyacinthaceae => Asparagaceae; Taccaceae => Dioscoreaceae; Sparganiaceae => Typhaceae; and Centrolepidaceae => Restioniaceae. On the other hand, some families sensu APG IV were newly established or circumscribed based on the emergence of some related genera of the other families in IFV and CPSV, such as *Acorus* (Araceae) was transferred to Acoraceae; *Tenagocharis* and *Limnocharis* (Butomaceae) => Alismataceae; *Petrosavia* (Melianthaceae in IFV or Liliaceae in CPSV) => Petrosaviaceae; *Paris* (Liliaceae in IFV) => Melanthiaceae; *Disporum, Gloriosa*, and *Iphigenia* (Liliaceae in IFV or Melanthiaceae and Convallariaceae in CPSV) => Colchicaceae; *Curculigo* and *Hypoxis* (Amaryllidaceae) => Hypoxidaceae; *Agave, Dracaena, Sansevieria, Urginea* (Agavaceae in IFV), *Asparagus*, *Aspidistra, Disporopsis, Disporum, Polygonatum, Tupistra* (Liliaceae in IFV), and *Nolina* (Agavaceae in IFV or Nolinaceae in CPSV) => Asparagaceae; and *Costus* (Zingiberaceae) => Costaceae. Therefore, the monocots in Vietnam include 76 families and 10 orders, according to APG IV, of which 3 families (Acoraceae, Melanthiaceae, and Petrosaviaceae) and 4 orders (Acorales, Asparagales, Dioscoreales, and Petrosaviales) were newly circumscribed for Flora of Vietnam (Appendix A).

*Ceratophyllales*: The independent lineage sensu APG IV containing only one order, Ceratophyllales, is the probable sister of eudicots. The order Ceratophyllales has only one family, Ceratophyllaceae, which was also recognized in both IFV and CPSV. Still, it was placed in the order Nymphaeales, one member of the ANA group sensu APG IV (Appendix A).

*Eudicots*: The eudicots in Vietnam sensu IFV includes 179 families, while CPSV recognized 203 families in Vietnam. The number of families of the eudicots sensu IFV and CPSV is more than those in APG IV, which recognized 155 families. Among the recognized families of the eudicots in Vietnam, 93 families in IFV and CPSV have the same delimitation as APG IV (Appendix A). Some families sensu IFV and CPSV were merged into the other families in APG IV such as: Leeceae was merged into Vitaceae; Rhoipteleaceae => Juglandaceae; Sịphonodontataceae => Celastraceae; Turneraceae => Passifloraceae; Hugoniaceae => Linaceae; Cochlospermaceae => Bixaceae; Erythropalaceae => Olacaceae; Viscaceae => Santalaceae; Chenopodiaceae => Amaranthaceae; Tetragoniaceae => Aizoaceae; Myrsinaceae => Primulaceae; Epacridaceae, Pyrolaceae => Ericaceae; Potaliaceae => Gentianaceae; Asclepiadaceae => Apocynaceae; Cuscutaceae => Convolvulaceae; Callitrichiaceae => Plantaginaceae; and Myoporaceae => Buddlejaceae. Furthermore, some families sensu APG IV were newly established or circumscribed based on the emergence of some related families in IFV and CPSV, e.g., Dipsacaceae and Valerianaceae, part of Caprifoliaceae (*Abelia, Lonicera*) were merged into Caprifoliaceae; Epacridaceae and Pyrolaceae in CPSV => Ericaceae; Punicaceae, Sonneratiaceae, and Trapaceae => Lythraceae; Bombacaceae, Sterculiaceae, and Tiliaceae => Malvaceae; part of Cornaceae (*Mastixia*) in IFV and Mastixiaceae and Davidiaceae in CPSV => Nyssaceae; Aceraceae and Hippocastanaceae => Sapindaceae; Buddlejaceae and Myoporaceae => Scrophulariaceae; Ceasalpiniaceae and Mimosaceae => Fabaceae; and Epacridaceae and Pyrolaceae in CPSV => Ericaceae.

Some families sensu APG IV were also newly established or circumscribed based on the merging of some related genera of the other families in IFV and CPSV, such as *Gynocardia* and *Hydnocarpus* (Flacourtiaceae) were transferred to Achariaceae; *Hydrocotyle* (Apiaceae) => Araliaceae; *Calophyllym* and *Mesua* (Clusiaceae) => Calophyllaceae; *Pentaphragma* (Campanulaceae) => Pentaphragmataceae; *Gonocaryum* (Icacinaceae) => Cardiopteridaceae; *Carlemannia* (Rubiaceae) and *Silvianthus* (Caprifoliaceae) => Carlemanniaceae; *Chrysobalanus* and *Parinari* (Rosaceae in IFV) => Chrysobalanaceae; *Polyosma* (Iteaceae in IFV or Escallonoiaceae in CPSV) => Escallonoiaceae; *Aucuba* (Cornaceae in IFV or Acubaceae in CPSV) => Garryaceae; *Gisekia* (Aizoaceae in IFV or Molluginaceae in CPSV) => Gisekiaceae; *Cratoxylum* and *Hypericum* (Clusiaceae in IFV) => Hypericaceae; *Callicarpa, Caryopteris, Clerodendron, Congea, Garrettia, Gmelina, Hymenopyramis, Karomia, Prema, Schnabelia, Sphenodesme, Tectona, Teijsmanniodendron, Tsoongia,* and *Vitex* (Verbenaceae) => Lamiaceae; *Hugonia* and *Indoroucheria* (Sabiaceae) => Linaceae; *Legazpia, Linderna, Picria, Pierranthus,* and *Torenia* (Scrophulariaceae) => Linderniaceae; *Pilostigma* (Saxifragaceae) => Loranthaceae; Mazus (Scrophulariacea) => Mazaceae; *Mitrastemon* (Rafflesiaceae) => Mitrastemonaceae; *Mastixia* (Cornaceae in IFV) => Nyssaceae; *Erythropalum* (Oleaceae in IFV or Erythropalaceae in CPSV) => Olacaceae; *Alectra, Brandisia, Buchnera, Centranthera, Lindendergia, Pedicularis, Rehmannia, Sopubia, Striga,* and *Wightia* (Scrophulariaceae) => Orobanchaceae; *Paulownia* (Scrophulariaceae) => Paulowniaceae; *Adinandra, Anneslea, Eurya,* and *Ternstroemia* (Theaceae) => Pentaphylacaceae; *Penthorum* (Pittosporaceae in IFV) => Penthoraceae; *Mimulus, Microcarpaea* and *Glossotigma* (Scrophulariaceae) => Phrymaceae; *Adenosma, Angelonia, Antirrhinum, Bacopa, Digitalis, Limnophila, Russelia, Scoparia, Stemodia,* and *Veronica* (Scrophulariaceae) => Plantaginaceae; *Drypetes* (Euphorbiaceae) => Putranjivaceae; *Neothorelia, Stixis,* and *Tirania* (Capparaceae) => Resedaceae; *Harrisonia* (Simaroubaceae) => Rutaceae; *Ginalloa, Korthalsella,* and *Viscum* (Loranthaceae in IFV) => Santalaceae; *Schoepfia* (Olacaceae in IFV) => Schoepfiaceae; *Gomphandra* (Icacinaceae) => Stemonuraceae; *Suriana* (Simaroubaceae) => Surianaceae; *Talinum* (Portulacaceae) => Talinaceae; *Tapiscia* (Staphyleaceae) => Tapisciaceae; and *Torricellia* (Cornaceae in IFV) => Torricelliaceae (Appendix A).

Notably, the delimitation of families of eudicots in Vietnam sensu IFV and CPSV has remarkably changed since APG III [7] and APG IV [8], e.g., Euphorbiaceae, Saxifragaceae, Scrophulariaceae, Capparaceae, Flacourtiaceae, Verbenaceae, and Cornaceae sensu IFV and CPSV were divided into some different families in APG IV. Euphorbiaceae sensu IFV and CPSV were divided into Euphorbiaceae s.s. Phyllanthaceae and Putranjivaceae in APG IV. Saxifragaceae sensu IFV and CPSV were divided into Loranthaceae, Hydrangeaceae, and Saxifragaceae s.s. in APG IV. Scrophulariaceae sensu IFV and CPSV were divided into Plantaginaceae, Linderniaceae, Mazaceae, Phrymaceae, Paulowniaceae, Orobanchaceae, and Scrophulariaceae s.s in APG IV. Capparaceae sensu IFV and CPSV were divided into Resedaceae, Cleomaceae, and Capparaceae s.s. in APG IV. In APG IV, Flacourtiaceae sensu IFV and CPSV were divided into Achariaceae, Salicaceae, and Flacourtiaceae s.s. in APG IV. Verbenaceae sensu IFV and CPSV were divided into Lamiaceae, Acanthaceae, and Verbenaceae s.s. in APG IV. Especially, Cornaceae sensu IFV was divided into Aucubaceae, Mastixiaceae, Torricelliaceae, Nyssaceae, and Cornaceae s.str. in CPSV, while APG IV merged Aucubaceae in Garryaceae and Mastixiaceae in Nyssaceae (Appendix A).

So far, the eudicots in Vietnam have been classified into 37 orders and 155 families according to APG IV, of which 18 families (Achariaceae, Adoxaceae, Akaniaceae, Caryophyllaceae, Carlemanniaceae, Cleomaceae, Garryaceae, Gisekiaceae, Mitrastemonaceae, Linderniaceae, Paulowniaceae, Pentaphragmataceae, Phyllanthaceae, Resedaceae, Stemonuraceae, Surianaceae, Tapisciaceae, and Talinaceae) and 16 orders (Apiales, Aquifoliales, Boraginales, Brassicales, Caryophyllales, Crossosomatales, Escalloniales, Garryales, Geraniales, Huerteales, Malpighiales, Myrtales, Oxalidales, Santalales, Solanales, and Zygophyllales), were newly circumscribed for Flora of Vietnam.

## 3. Materials and Methods

The delimitations of IFV families followed the last edition of IFV volumes 1–3 [17]. Those of the CPSV followed CPSV volumes 1–3 [2]. 

The delimitation of families and assignment of genera of lycophytes and ferns and gymnosperms in Vietnam is mainly based on PPG I [9] and GPG [10], respectively. However, the assignment of genera of angiosperms was often not explicit in APG IV [8], which is primarily a list of accepted families and the orders into which they are classified, and thus, re-delimitations of many of the families were not based on APG IV alone. Therefore, the consultation of multiple sources of information (e.g., recent phylogenetic studies, description of genera and families, and taxonomic revisions) was necessary. Furthermore, the Angiosperm Phylogeny Website (APweb) [62] played an important role in deciding how genera of angiosperms were placed.

## 4. Conclusions

IFV classified lycophytes and ferns of Vietnam in 139 genera and 33 families, gymnosperms in 25 genera and 10 families, and angiosperms in 2110 genera and 241 families. In contrast, CPSV recognized 143 genera and 34 families for lycophytes and ferns, 22 genera and 9 for gymnosperms, and 2210 genera and 265 for angiosperms of Vietnam. The classification of vascular plants in Vietnam in these previous treatments (IFV and CPSV) and the recent modern classifications (PPG I, “GCG”, and APG IV) are significantly different.

According to PPG I classification, the lycophytes and ferns of Vietnam were classified into 134 genera, 37 families, 14 orders, 3 subclasses, and 2 classes, in which, 6 families were newly circumscribed for Vietnam. The gymnosperms of Vietnam were classified into 23 genera, 8 families, 6 orders, and 4 subclasses, according to the GPG classification, of which the delimitation of Cupressaceae and Taxaceae sensu GPG remarkably changed in comparison with those in IFV and CPSV. According to APG IV, the angiosperms of Vietnam were classified into 246 families and 54 orders. Compared with IFV and CPSV, the delimitation of many genera and families of angiosperms of Vietnam following APG IV significantly changed, and 21 families and 23 orders were newly circumscribed for flora of Vietnam.

## Figures and Tables

**Figure 1 plants-12-00967-f001:**
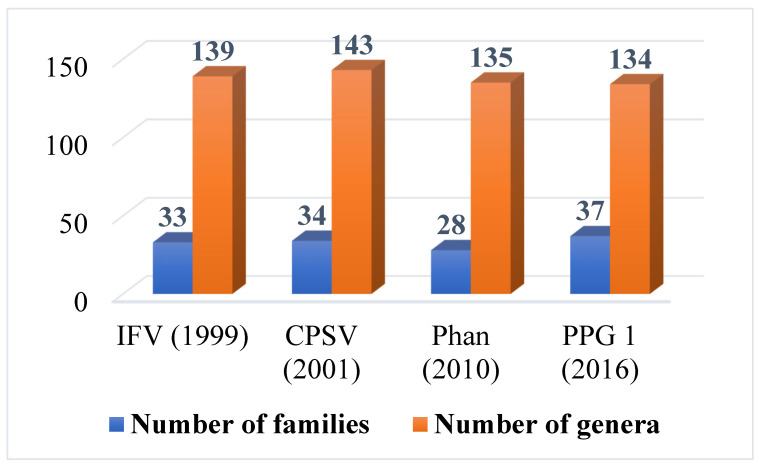
Number of families and genera of lycophytes and ferns in Vietnam according to different treatments.

**Figure 2 plants-12-00967-f002:**
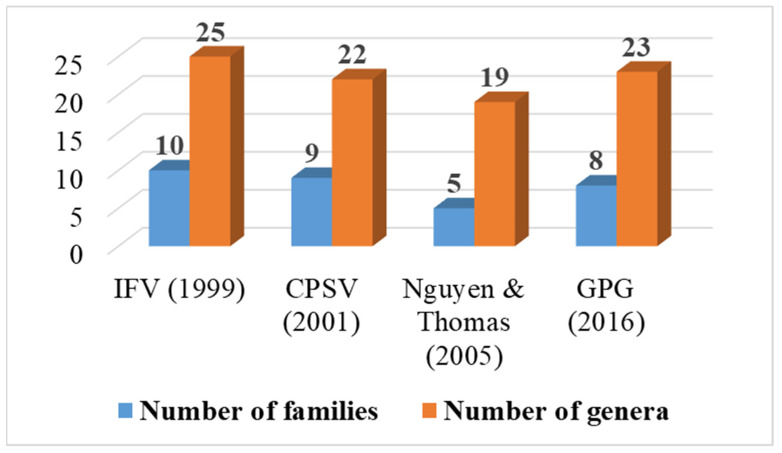
Number of families and genera of gymnosperms in Vietnam according to different treatments.

**Figure 3 plants-12-00967-f003:**
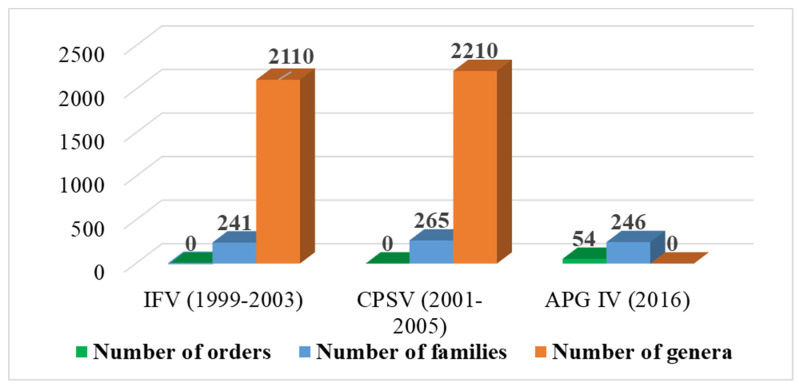
Number of orders, families, and genera of angiosperms in Vietnam according to different treatments.

## Data Availability

The study did not report any additional data.

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
