# Peer review of "Classification of Vascular Plants in Vietnam According to Modern Classification Systems"

_plants, 2023, doi:10.3390/plants12040967_

Round 1
Reviewer 1 Report
Dear Editor and Authors,
I have read the current MS which is quite interesting and possible to publish in Plants journal. The authors are well-written in whole texts. In addition, there is much valuable information in supp. materials. In my opinion, all separated supp. materials are combined into a single pdf file which is better to see by other researchers.
Author Response
We greatly appreciate your positive comments and suggestions on our manuscript. However, we think that the supplemental materials should be separated from each other so that the readers can easily follow them.
Reviewer 2 Report
Thanks for considering Plants for your research manuscript.
I have a few notes that I listed below:
In section 3:
you should use the term Ho Chi Minh City and not Saigon at the first is the official name of the city.
There is an extension discussion about the change in families from the IFV and APGIV with the change in family classification as 'Centrolepidaceae => Restioniaceae' for example. This discussion goes for over one page. Indeed is important, however I found it difficult to follow.
Change the last two sentence of the conclusion. '...not only for botanists, ecologists, conservationists, and medical re- searchers of Vietnam but also for those international botanists. They are interested in Asian botany in general and the Flora of Vietnam.'
I think the last sentence or is merged to the previous sentence or needs to be cancelled.
The graphs need to be improved. I think you have used exl to complete them but they are estetically not nice and with a poor standard for a journal which has an high IF.
I would suggest to improve both of them. If you are not familiat with R and ggplots there are online software which allows you to produce histogram with a better layout.
Author Response
Rebuttal to the comments by reviewer (answering in blue)
I have a few notes that I listed below:
In section 3: you should use the term Ho Chi Minh City and not Saigon at the first is the official name of the city.
Rewritten
There is an extension discussion about the change in families from the IFV and APGIV with the change in family classification as 'Centrolepidaceae => Restioniaceae' for example. This discussion goes for over one page. Indeed is important, however I found it difficult to follow.
This discussion is a very important part of the manuscript which show the changes in family classification from IFV and APG IV. We think that the current presentation is the best way to highlight which were changed.
Change the last two sentence of the conclusion. '...not only for botanists, ecologists, conservationists, and medical re- searchers of Vietnam but also for those international botanists. They are interested in Asian botany in general and the Flora of Vietnam.' I think the last sentence or is merged to the previous sentence or needs to be cancelled.
This paragraph was removed
The graphs need to be improved. I think you have used exl to complete them but they are estetically not nice and with a poor standard for a journal which has an high IF. I would suggest to improve both of them. If you are not familiat with R and ggplots there are online software which allows you to produce histogram with a better layout.
Many thanks for your positive comments and suggestions. We think that Microsoft Excel is more than enough to produce the simple graphs in this case. By the way, in this revision, we have improved these graphs and getting a better layout.
Reviewer 3 Report
The reviewed manuscript is dedicated to the fit between floristic records of Vietnam and the most contemporary plant systematics.
When reviewing and rejecting a paper, I often suggest a set of changes (sometimes quite profound) which can be done to improve a paper, so the resulting decision is more like 'reject with an encouragement to resubmit'. However in this particular case I cannot conclude like this because the whole concept of such comparison seems of little significance for me.
The key problem which persists through the whole work is that phylogeny and taxonomy cannot be limited by countries' boarders nor by floristic regions. For example, if some floristic list contains Cassia (Caesalpiniaceae), the same record can be updated to Senna (Fabaceae: Caesalpinioideae), and it is of no importance whether this particular senna grows in Spain, India or Vietnam. When authors introduce the section 'History of classification of vascular plants in Vietnam', they describe the history of floristic studies in Vietnam instead. Authors themselves state that 'the angiosperms in Vietnam [were classified] following Takhtajan classification system'. Any works on the Vietnamese flora can follow systems of De Candolle, Takhtajan or the APGIV, but these and other systems exist independently from region.
By the way, in my opinion, only this brief review of history of floristic studies in Vietnam deserves a publication, as it really represents some data which can be interesting for scientific community worldwide.
The key product of authors' work, i.e. big tables matching different systems, would be hardly of any use in contemporary world. If one decides to check which family the exact herbarium specimen belongs to now, he/she will simply use the APGIV itself or web-searchable (and continuously updated) databases like The Plant List, World Flora Online or the Angiosperm Phylogeny Website.
These are the major flaws of the reviewed work which, in my opinion, make it unacceptable for publication in Plants.
As for minor issues, some of them are available in the manuscript file itself (see attached). There are some statements which at the moment sound ambiguously and require rephrasing (e.g. 'DNA molecular technique'). In the supplementary tables, there are numerous spelling mistakes such as Ceasalpiniaceae or Avincennia which will probably complicate the use of these tables even more.
To sum up, I recommend to reject this paper.

Author Response
Rebuttal to the comments by reviewer (answering in blue)
Reviewer 3
The reviewed manuscript is dedicated to the fit between floristic records of Vietnam and the most contemporary plant systematics.
When reviewing and rejecting a paper, I often suggest a set of changes (sometimes quite profound) which can be done to improve a paper, so the resulting decision is more like 'reject with an encouragement to resubmit'. However in this particular case I cannot conclude like this because the whole concept of such comparison seems of little significance for me.
The key problem which persists through the whole work is that phylogeny and taxonomy cannot be limited by countries' boarders nor by floristic regions. For example, if some floristic list contains Cassia (Caesalpiniaceae), the same record can be updated to Senna (Fabaceae: Caesalpinioideae), and it is of no importance whether this particular senna grows in Spain, India or Vietnam. When authors introduce the section 'History of classification of vascular plants in Vietnam', they describe the history of floristic studies in Vietnam instead. Authors themselves state that 'the angiosperms in Vietnam [were classified] following Takhtajan classification system'. Any works on the Vietnamese flora can follow systems of De Candolle, Takhtajan or the APGIV, but these and other systems exist independently from region.
By the way, in my opinion, only this brief review of history of floristic studies in Vietnam deserves a publication, as it really represents some data which can be interesting for scientific community worldwide.
The key product of authors' work, i.e. big tables matching different systems, would be hardly of any use in contemporary world. If one decides to check which family the exact herbarium specimen belongs to now, he/she will simply use the APGIV itself or web-searchable (and continuously updated) databases like The Plant List, World Flora Online or the Angiosperm Phylogeny Website.
These are the major flaws of the reviewed work which, in my opinion, make it unacceptable for publication in Plants.
As for minor issues, some of them are available in the manuscript file itself (see attached). There are some statements which at the moment sound ambiguously and require rephrasing (e.g. 'DNA molecular technique'). In the supplementary tables, there are numerous spelling mistakes such as Ceasalpiniaceae or Avincennia which will probably complicate the use of these tables even more.
To sum up, I recommend to reject this paper.
We greatly appreciate your positive comments and suggestions on our manuscript. As you know that classical classification systems, which were established mainly based on morphological data that did not reflect evolutionary history and, ultimately led to instability of natural groups (genera and families). Therefore, the position of a genus or a family might be inconsistent in the different classical classification systems by the different authors even the same author. The limitation of classical classification systems led to challenges and difficulties in arranging the herbarium collection, specimen exchange, and further studies of the genetic diversity.
Nowadays, the surge in the application of molecular biology techniques has resulted in significant contributions to plant systematics, evolutionary questions, and phylogeny. As a result, various DNA-based modern classification systems were established, e.g., APG (the Angiosperm Phylogeny Group classification) for flowering plants, PPG I (the Pteridophyte Phylogeny Group classification) for lycophytes and ferns, “GPG” (the Gymnosperm Phylogeny Group classification) for gymnosperms. These classifications present the linear sequence and delimitation of genera and families of vascular plants recognizing their natural groups based on both morphological and molecular data analyses, utilizing a democratic community-based approach, which is currently widely accepted by scientists and herbaria around the world.
Actually, most Vietnamse botanists are still classifying the plant species according to the classical classifications without update of the newly circumscribed delimitation of genera and families. Therefore, we think that this current study is necessary and the results would be the foundation stones and helpful for future research on the flora of Vietnam, e.g. biodiversity assessment, the arrangement of plant collections in Vietnamese herbaria, specimen exchange based on the updated morden classifications
Rebuttal to the comments provided on the manuscript and copied here (answering in blue)
[Comment 1] Compared with what?
Rephrased
[Comment 2] Genetic diversity is a phenomenon which occurs without any contributions from human beings (probably except for negative influence). In the other words, people/scientists can contribute to systematics (which is a fruit of human mind) but not to diversity, only to the studies of diversity.
Rephrased
[Comment 3] I cannot understand what it means.
Rephrased
[Comment 4] Here and further: if you provide a reference in square brackets (e.g. [7]), there is no need to provide a year of publication.
Corrected here and through the manuscript
[Comment 5] I cannot understand what you mean by sequence of genera/families.
Rephrased
[Comment 6] It is very difficult to understand this sentence.
If I got your message right, you suggest that plants adapted to similar conditions are more likely to possess a similar set of metabolite (potentially those of medicinal value). It sounds quite strange and unrelated to the real situation with plant physiology.
This paragraph was removed
[Comment 7] sequence?
Rephrased
[Comment 8] It is an overgeneralization. As you may see, some of families did not change their content even after application of molecular phylogeny (e.g. Leguminosae). The key profit of molecular phylogeny is to distinguish between true relation and homoplasy, when morphological data confuses such delimitation.
We totally agree that some of vascular plant families di not change their content after application of molecular phylogeny that was well done, but the delimitation of many genera and families were changed (combined or splited) based on the recent molecular phylogenetic studies when morphological data confuses such delimitation.
[Comment 9] It is a principal flaw of your work, as it is the completely incorrect statement. Any of listed systems (e.g. APGIV) are actual regardless of floristic region or territory. Former Aceraceae are considered a member of Sapindaceae regardless of whether maples grow in Canada or in China.
We mean that the genera and families of vascular plants of Vietnam have not yet been classified and arranged according to modern classification systems
[Comment 10] With exception of fully endemic plant families, any other works on classification are independent from region.
Rephrased
[Comment 11] According to the Plants' guide for authors, the M&M section should be placed after Results and Discussion.
Corrected
[Comment 12] Floristic studies are generally unrelated to phylogeny.
Rephrased
[Comment 13] Format all references according to a journal's guide.
Rephrased
[Comment 14] In histogram itself, replace 'family' with 'families' ang 'genus' with 'genera'. This needs to be done in all other figures. Actually, I see no reason in figures like this, as all these data are available in the text itself.
Rephrased. Although the data are available in the text, the graphs is still necessary which woulld be help the readers to easily follow
[Comment 15] I don't think it is worth redescribing the APGIV system here. All these details can be found in the original paper.
This paragraph was removed
[Comment 16] I cannot understand why there are no genera in APGIV. If this system does not go below family level, don't provide such a metrics in this graph, as it is meaningless.
In APG IV (2016), only classification of orders and families of Angiosperms is presented. It does not go below family level (subfamily and genera in detail). However, we used the Angiosperm Phylogeny Website (Apweb) as an important role in deciding how genera of each family in angiosperms were placed.
[Comment 17] It is not a conclusion in its original sense but simply a repeat of what was already described in Results and Discussions sections.
Rephrased to highlight the main results from the current study
Round 2
Reviewer 2 Report
Please check my previous email